# Case Series Using Salvage Haplo-Identical Stem Cells for Secondary Transplantation

**DOI:** 10.3390/medicina59061077

**Published:** 2023-06-02

**Authors:** Alexandra Ionete, Zsofia Varady, Orsolya Szegedi, Daniel Coriu

**Affiliations:** 1Fundeni Clinical Institute, 022328 Bucharest, Romania; 2Faculty of General Medicine, University of Medicine and Pharmacy “Carol Davila”, 020021 Bucharest, Romania

**Keywords:** haploidentical transplant, bone marrow transplant, rejection of stem cell graft

## Abstract

In order to expand the donor pool and accessibility of the transplant procedure, it was necessary to introduce haplo-identical stem cell transplants in the Fundeni Clinical Institute from 2015. Even if the Romanian population is an ethnically compact white population, many of the patients referred for bone marrow transplant lack a suitable donor. Hematopoietic stem cell transplant from a haplo-identical donor is an alternative option for those patients without an HLA (Human Leucocyte Antigen)-matched donor (sibling or matched unrelated). This procedure was used also as a salvage option for those who experienced engraftment failure or the rejection of the first stem cell graft. In this case series, we present three such cases, with a haplo-transplant used as a salvage protocol (after an engraftment failure or rejection of the first transplanted cells). The patients we present were diagnosed with AML (acute myeloid leukemia) with MDS (myelodysplastic syndrome), MDS—RAEB 2 (myelodysplastic syndrome—refractory anemia with excess blasts 2), and SAA (severe aplastic anemia). In two of the three cases, the engraftment failure may have been due to the conditioning Fludarabine/Busulfan/Cyclophosphamide (Flu/Bu/CFA) used, combined with marrow grafts. In all three cases, the second transplant was of haplo-identical peripheral blood stem cells using Melphalan/Fludarabine (Mel/Flu) conditioning, the cells engrafted properly and the patients experienced complete chimerism, and two of them are alive with an excellent quality of life.

## 1. Introduction

Even if the Romanian population is an ethnically compact white population, many of the patients referred for bone marrow transplant lack a suitable donor.

The chance of finding a suitable HLA-matched donor is about 60–70% in the white population and <10% for different minorities [1,2].

The medium time for finding a suitable unrelated donor, from the initiation of searching until the donation of stem cells, is approximately 4 months; some patients do not have this time (4 months is too long, for example, in patients with acute leukemia) [3].

Hematopoietic stem cell (HSC) transplant from an matched unrelated donor using myeloablative conditioning (MAC) still has a high transplant mortality rate and a high long-term morbidity [4,5,6,7,8].

More than 95% of patients in need of an allogeneic stem cell transplant have a haplo-identical donor [9].

Haplo-identical stem cell transplant is associated with a high incidence of graft-versus-host disease (GVHD), late engraftment and graft failure [10,11]. GVHD and graft failure, the initial immunological barriers to haplo-hematopoietic stem cell transplant, have been overcome with different methods that control T cell alloreactivity post-transplant.

The T cell depletion of the graft leads to an imbalance between the host and donor T cells and thus results in high rates of graft failure. This imbalance may be overcome with the use of “megadose” stem cell grafts (e.g., >10 × 10^6^ CD34 + cells/kg and ≤4 × 10^4^/kg T cells in patients with severe SCID (combined immunodeficiency syndrome) [12,13].

There are two methods for T-cell depletion:-in vitro, using a CliniMACS device, in which the graft is depleted by both CD3 + cells (T lymphocytes) and CD19 + cells (B lymphocytes). For these cases, engraftment will appear at a CD34 + cell dose of 5.2 × 10^6^/kg. The graft is depleted by the CD19 + cells (B lymphocytes) in order to decrease the risk of developing lympho-proliferative diseases after transplant (PLDL) [14] and GVHD [15]-in vivo, using antithymocite globulin (ATG). ATG affects the host T lymphocytes and facilitates engrafting. ATG acts on donor T lymphocytes with effects on GVHD and post-transplant immunity.

One strategy for achieving the in vivo attenuation of T cell alloreactivity is the use of high-dose post-transplant cyclophosphamide (PTCy) [16].

PTCy may be used in reduced intensity conditioning (RIC) as well as in MAC conditioning. Its use leads to a decrease in the development of acute and chronic GVHD [16,17]. MAC conditioning with PTCy decreases the risk of relapse and increases non-relapse mortality (NRM) [18].

PTCy has better results than ATG in patients with acute myeloid leukemia (AML) [19,20].

In order to expand the donor pool and the accessibility of the transplant procedure, it was necessary to introduce haplo-identical stem cell transplants in the Fundeni Clinical Institute (ICF) from 2015. The Bone Marrow Transplant Unit of the ICF is the only place in Romania that performs haplo-transplant procedures.

Hematopoietic stem cell transplant from a haplo-identical donor is an alternative option for those patients without a human leukocyte antigen (HLA)-matched donor, sibling or unrelated, so it was included in our transplant strategy.

This procedure can be also used as a salvage option for those who have experienced engraftment failure or a rejection of the first stem cell graft. In this case series, we would like to present three such cases, with haplo-transplant used as a salvage protocol.

## 2. Materials and Methods

Case presentation: the first three cases (in Romanian health system) undergoing a secondary haploidentical stem cells transplant, used for salvage after engraftment failure or the rejection of the first transplanted cells [21,22,23].

### 2.1. The First Case

The patient is a 28-year-old male diagnosed with AML/MDS, transplanted, and in first complete remission (CR1), with his sister as the haplo-identical donor (5/10 HLA match); this was due to the unavailability of a matched unrelated donor within the registry in a timely manner.

The first transplant was conditioned with the following (Flu/CFA/BU):-Fludarabine (Flu): 30 mg/m^2^/day × 5 days (day - 6, day - 5, day - 4, day - 3, day - 2);-Cyclophosphamide (CFA): 14.5 mg/kg bw/day × 2 days (day - 6, day - 5);-Busulfan (BU): 3.2 mg/kg bw × 2 days (day - 3, day - 2).

The GVHD prophylaxis used the following (PTCy + tacro + MMF):-PTCy (day + 3 and day + 4);-Tacrolimus (tacro): 0.12 mg/kg/day, divided in two doses from day + 5;-Mycophenolat mofetil (MMF): 15 mg/kg/dose in 2–3 doses/day from day + 5 [24,25,26,27,28,29,30,31,32,33,34,34,35,36,37,38].

The stem cells we used were harvested from the marrow (0.97 × 10^6^ CD34+/kgbody weight) of the donor on day 1. For the patient, we used G-CSF (granulocyte colony-stimulating factor) at a dose of 300 micrograms/day from day + 5. The patient did not engraft at all, so we had to use a salvage protocol; because of the the absence of a new donor, we repeated the haplo approach from the same haploidentical sister. However, the second time, we changed the conditioning and the source of the cell graft as well.

For the second conditioning we used (MEL/Flu):-Melfalan (MEL): 140 mg/m^2^/day on day - 6-Fludarabine (Flu): 40 mg/m^2^/day on day - 5, day - 4, day - 3 and day - 2

The GVHD prophylaxis was the same as that in the first setting: PTCy + tacro+ MMF.

The cell graft this time was harvested from mobilized peripheral blood using 4 days of mobilization for the haplo donor with G-CSF at 10 MU/kg/day. The peripheral stem cell graft was 4.4 × 10^6^ CD34+/kg of body weight. He engrafted on day + 15, with complete chimera on day + 30, + 60, + 90 at 6, 9 and 12 months. Immune suppression was stopped at 5 months. (See Table 1).

### 2.2. The Second Case

The second patient was a 37-year-old female diagnosed with myelodysplastic syndrome—refractory anemia with excess blasts 2 (MDS-AREB2), transplanted, and in second complete remission (CR2), with her haplo-identical uncle with 5/10 HLA match as the donor (no siblings, no matched unrelated donors, no parents, and one little child for whom the mother/patient refused general anesthesia and the use of G-CSF to mobilize).

The first transplant was conditioned with the following (Flu/CFA/BU):-Fludarabine: 30 mg/m^2^/day × 5 days (day - 6, day - 5, day - 4, day - 3, day - 2);-Cyclophosphamide: 14.5 mg/kg body weight/day × 2 days (day - 6, day - 5);-Busulfan: 3.2 mg/kg body weight × 2 days (day - 3, day - 2).

GVHD prophylaxis was as follows (PTCy + tacro + MMF):-PTCy: (day + 3 and day + 4);-Tacrolimus: (0.12 mg/kg/day divided into two doses from day + 5);-Mycophenolat mofetil: 15 mg/kg/dose in 2–3 doses/day from day +5 [16,39,40,41,42,43,44].

The first graft we used was marrow (2.17 × 10^6^ CD34+/kg body weight; 2.8 × 10^8^ TNC <total nucleated cells>/kg body weight) with G-CSF from day + 5. The patient was engrafted and the chimerism on day +30 was fully the donor’s. Shortly after, she experienced rejection, with a chimeric profile on day + 48 of only 3% donor cells. We had to re-transplant her with the same haplo-identical uncle in the absence of another donor.

The second conditioning was as follows (MEL/Flu):-Melfalan: 140 mg/m^2^/day on day 6;-Fludarabine: 40 mg/m^2^/day in day - 5, day - 4, day - 3 and day - 2.

The GVHD prophylaxis was PTCy + tacro + MMF.

The cell graft this time used peripheral stem cells, at 4.4 × 10^6^ CD34+/kgbw, and we used G-CSF from day + 5. She engrafted on day + 16, with complete chimera on day + 30. She died as a result of broncho-pneumonia and multiorgan failure on day + 42 after the second transplant. (See Table 1).

### 2.3. The Third Case

The third patient was a 19-year-old male with severe aplastic anemia (SAA), diagnosed in January 2015, with a good initial response to immune-suppression therapy. He was transplanted after a relapse in 24 February 2016 with cells from a 10/10 HLA female; she was a matched unrelated donor (MUD) with the same blood group and same positive CMV status.

The conditioning protocol for the MUD transplant was the EBMT protocol (Flu/CFA):-Fludarabine: 30 mg/m^2^/day in day - 5, - 4, - 3 and - 2;-Cyclophosphamide: 300 mg/m^2^ in day - 5, - 4, - 3 and - 2.

The GVHD prophylaxis was as follows (CsA + MTX + ATG):-CsA (Cyclosporine A) from day - 1;-MTX on day + 1, + 3 and + 6;-ATG (Antithymocyte globulin): 10 mg/kg/day on day - 4, - 3, - 2 and - 1.

The cell graft this time was bone marrow: 3.1 × 10^6^ CD34+/kg bw [45].

He was engrafted on day + 23 with G-CSF. The chimeric profile was 100% donor cells on day + 30, and 84% donor cells at 3 months. We stopped the immune suppression at 5 months, but the chimerism decreased continuously. At 6 months, he had 23% donor chimera and the CMV and parvo-virus were reactivated also. His body totally rejected his graft with these complications.

On 23 September 2016 (7 months after the female MUD transplant), he received peripheral blood stem cells from his haplo-identical sister (5/10 HLA match) using 5.23 × 10^6^ CD34+/kg of body weight.

The conditioning protocol was Mel/Flu, as in the first two cases of re-transplant.

The GVHD prophylaxis was PTCy + MMF + tacro.

The engraftment was on day + 24 with G-CSF. The chimeric profile was 100% donor on day + 30, + 60 and + 90. He only experienced a CMV reactivation and one bacterial infection with Enterococcus sp after the transplant.

He is alive and well. (See Table 1).

## 3. Results

See Table 1.

**Table 1 medicina-59-01077-t001:** The characteristics of the three cases.

	Patient 1	Patient 2	Patient 3
Age (years)	28	37	19
Gender	Male	Female	Male
Diagnosis	AML with MDS	MDS—RAEB 2	SAA
Illness status	CR1	CR2	-
First transplant date	05 November 2015	22 January 2016	24 February 2016
First transplant donor	Haploidentical sister (5/10)	Haploidentical uncle (5/10)	10/10 MUD female donor
First graft	Marrow: 0.97 × 10^6^ CD34+/kg bw	Marrow: 2.17 × 10^6^ CD34+/kg bw	Marrow: 3.1 × 10^6^ CD34+/kg bw
First transplant conditioning and GVHD prophylaxis	Flu/BU/CFA with PTCy + tacro+ MMF	Flu/BU/CFA with PTCy + tacro+ MMF	Flu/CFA with CsA + MTX 1, 3, 6 and ATG_F 40 mg/kg bw total dose
First transplant blood group matching	Both A Rh positive	Recipient AB negative, donor A negative	Both A positive
First transplant CMV status in the pair	Both CMV positive	Both CMV positive	Both CMV positive
Engraftment of the first transplant	No engraftment with G-CSF (chimeric profile on day + 30: 0% donor cells)	Engraftment in day + 25 with G-CSF (chimeric profile 100% donor at day + 30) but rejection until day + 48 (chimeric profile with 3% of donor cells)	Engraftment in day + 23 with G-CSF (chimeric profile 100% donor at day + 30) but 23% at 6 months with aplastic marrow
Complications in first transplant	Fever of unknown origin	Grade III mucositis, genital bleeding	Grade III mucositis and fever of unknown origin
Second transplant date	23 December 2015 (day + 48 from the first transplant)	23 March 2016 (day + 58 from the first transplant)	29 September 2016 (7 months after MUD transplant)
Second transplant donor	Same haploidentical sister	Same haploidentical uncle	Haploidentical sister (5/10)
Second graft	PBSC: 4.4 × 10^6^ CD34+/kg bw	PBSC: 10.6 × 10^6^ CD34+/kg bw	PBSC: 5.23 × 10^6^ CD34+/kg bw
Second transplant conditioning and GVHD prophylaxis	MEL/Flu with PTCy + tacro + MMF	MEL/Flu with PTCy + tacro+ MMF	MEL/Flu with PTCy + tacro+ MMF
Second transplant blood group matching	Both A Rh positive	Recipient AB negative, donor A negative	Both A Rh positive
Second transplant CMV status in the pair	Both CMV positive	Both CMV positive	Both CMV positive
Engraftment after the second transplant	Day + 15 with G-CSF with complete chimerism on day + 30	Day + 16 with G-CSF with complete donor chiemrism in day + 30	Day + 24 with G-CSF with complete donor chimerism on day + 30, + 60, + 90
Complications of the second transplant	Enterocacter cloacae sepsis and a nodular lesion of the lungs	Bronhopneumonia with multiorgan failure	Grade IV mucositis and fever of unknown origin
Actual patient status	Alive and perfect clinical status Complete chimera CMV reactivation	Death in day + 42 after the second transplant	Alive and perfect clinical status Complete chimera CMV reactivation

## 4. Discussion

In recent years, haplo-identical family members have become one of the valid choices for allogeneic stem cell transplant.

In Romania, the algorithm used to choose a donor for a patient starts with family HLA typing in order to achieve the following:-find a sibling donor as a first step;-additionally identify the haplotypes within the family in case a haplo-identical transplant is needed.

The second step is to look in international registries for a matched unrelated donor. In the case of a patient who does not have a sibling or a matched unrelated donor in a timely manner, the haplo-identical transplant is proposed.

We presented three cases using the haplo protocol in order to save the lives of patients who have experienced an engraftment failure or rejections. The engraftment failure may be due (in the first two cases) to the conditioning we used (Flu/Bu/CFA), combined with marrow grafts. The source of the hematopoietic stem cells is also an important factor in the failure of the hematopoietic stem cells (HSC) transplant. In all three cases, the amount of CD34+ in the marrow graft was poor (0.97 × 10^6^ CD34+ cells/kg recipient; 2.17 × 10^6^ CD34+ cells/kg recipient; 3.1 × 10^6^ CD34+ cells/kg recipient). It is well known that an insufficient quantity of stem cells in the graft can lead to engraftment failure. This small amount of stem cells from the bone marrow graft can be explained by the fact that, in all three cases, the weights of the recipients were significantly higher than the weights of the donors. In addition, in the first two cases, the donors had a 50% HLA match with the recipients.

For the second transplant, another type of RIC conditioning (Mel/Flu) and another source of stem cells (peripheral blood stem cells) were used in all three patients. The number of stem cells from the peripheral blood was superior compared to the that from the first transplant (4.4 vs. 0.97; 10.6 vs. 2.17, respectively, 5.23 vs. 3.1). The donor for the third patient was changed for the second transplant, unlike the first two patients. This patient had a MUD 10/10 HLA match for the first transplant, but for the second one, his haplo-identical sister with a 5/10 HLA match became the donor.

HLA-haploidentical stem cells post-transplant, with a high-dose cyclophosphamide, are associated with a low rate of GVHD and non-relapse mortality (as the Baltimore group has already demonstrated).

Both sources of stem cell grafts (BM, PBSC) are acceptable options at the present time. All three patients grafted when peripheral blood was used as the source of the stem cells graft. This is in accordance with the medical literature that shows superior rates of engraftment for patients who receive peripheral blood stem cells (PBSC):-Couban et al., 2002: superior rates of engraftment for AML patients who receive peripheral blood stem cells (PBSC). At the same time, these patients have a higher risk of acute and chronic graft versus host disease (GVHD) [46].-Anaseti et al., 2012: comparable survival rates and relapse rates in AML patients transplanted with PBSC MUD grafts vs. AML patients transplanted with bone marrow (BM) grafts. It is demonstrated that BM grafts have a higher rate of graft failure and PBSC grafts are accompanied by higher rates of chronic GVHD [47].-Schlenk et al., 2010: MSD (matched sibling donor) and MUD (matched unrelated donor) are the preferred options for HSC transplant. For patients who do not have these types of donors, haplo-identical donors can be found [48].

In all three cases, the second transplant involved haplo-identical peripheral blood stem cells using Mel/Flu conditioning; the cells engrafted properly and the patients became complete chimeras, and two of them are alive with a good quality of life.

## Data Availability

Data used in this study may be provided by the corresponding author upon reasonable request.

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
