# Peer review of "Case Series Using Salvage Haplo-Identical Stem Cells for Secondary Transplantation"

_medicina, 2023, doi:10.3390/medicina59061077_

Round 1
Reviewer 1 Report
Current manuscript discussing case studies, I have followings suggestions and comments:
It would have been better if authors would have given HLA match scores of recipient and donors.
Reasons given for the failure of first transplant is quite vague. Authors mentioned that it may be because of conditioning they used for first transplant like Flu/Bu/CFA along with bone marrow stem cells.
But in second transplant they used peripheral blood stem cells with reduced conditioning like Mel/Flu with significant success rate.
It is therefore important to discuss the reasons why first conditioning was detrimental than second?
Discussion is too weak. HLA-haploidentical hematopoietic stem cells post-transplant with high dose of cyclophosphamide is already known to be associated with low rate of GVHD and non-relapse mortality. It is quite safe. Baltimore group has already demonstrated.
Perhaps authors suggested that reduced intensity of conditioning is better as done in second transplant and safe compared to first transplant with high conditioning.
Manuscript need English language editing.
Manuscript need English language editing. Need correction of English spelling mistakes.
Author Response
Rewier 1 Comments
Point 1: It would have been better if authors would have given HLA match scores of recipient and donors.
Response 1: All the 3 haplo-transplants were 50% matched
Point 2: Reasons given for the failure of first transplant is quite vague. Authors mentioned that it may be because of conditioning they used for first transplant like Flu/Bu/CFA along with bone marrow stem cells.
But in second transplant they used peripheral blood stem cells with reduced conditioning like Mel/Flu with significant success rate.
It is therefore important to discuss the reasons why first conditioning was detrimental than second?
Discussion is too weak. HLA-haploidentical hematopoietic stem cells post-transplant with high dose of cyclophosphamide is already known to be associated with low rate of GVHD and non-relapse mortality. It is quite safe. Baltimore group has already demonstrated.
Perhaps authors suggested that reduced intensity of conditioning is better as done in second transplant and safe compared to first transplant with high conditioning.
Response 2: The source of hematopoietic stem cells is also an important factor for failure of the hematopoietic stem cells (HSC) transplant.
Couban et al 2002: superior rates of engraftement for AML patients who receive peripheral blood stem cells (PBSC). In the same time, these patients have a higher risk for acute and chronic graft versus host disease (GVHD)
Anaseti et al 2012: comparable survival rates and relapse rates in AML patients transplanted with PBSC MUD grafts vs. AML patients transplanted with bone marrow (BM) grafts. It is showed that BM grafts have a higher rates of graft failure and PBSC grafts are accompanied by higher rates of chronic GVHD.
Both sources of stem cells grafts (BM, PBSC) are acceptable options in present time.
Schlenk et al 2010: MSD (matched sibling donor) and MUD (matched unrelated donor) are the preferred option for HSC-transplant. For patients who do not have these types of donors can be appealed haplo-identical donors.
The source of hematopoietic stem cells is also an important factor for failure of the hematopoietic stem cells (HSC) transplant. In all three cases the amount of CD34+ in the marrow graft was poor (0,97 x 106 CD34+cells/ kg recipient; 2,17 x 106 CD34+cells/ kg recipient; 3,1 x 106 CD34+cells/ kg recipient). It is well known that an insufficient quantity of stem cells in the graft can lead to engraftment failure. This small amount of stem cells from the bone marrow graft can be explained by the fact that, in all three cases, the weights of recipients were significantly higher than the weights of the donors. In addition, in the first two cases donors had a 50% HLA match with recipients.
For the second transplant, in all 3 patients was used another type of RIC conditioning (Mel/Flu) and another source of stem cells (peripheral blood stem cells). The amount of stem cells from peripheral blood was superior compared to the one from the first transplant (4,4 vs 0,97; 10,6 vs 2,17 respectively 5,23 vs 3,1). The donor for the third patient was changed for the second transplant, unlike the first two patients. This patient had a MUD 10/10 HLA match for the first transplant but for the second one, his haplo-identical sister with a 5/10 HLA match became the donor.

Reviewer 2 Report
The manuscript is interesting and the results reported are attractive for the possible use of the hematopoietic stem cell transplantation from haplo-identical donor as valid and alternative option for patients without HLA-matched donor, however there are several open questions and a number of changes which are needed.
The text of the manuscript is confusing, it reports a lot of acronyms that makes the reading of the manuscript difficult for scientists of different disciplines who are not very familiar with this type of research. It must also be revised for English fluency and typing errors.
-Please define the acronyms at first usage and use them alternated together with the long name to make the manuscript more fluid and understandable.
Abstract:
-Acronyms not defined: Please specify all acronyms reported.
Introduction:
-Please revised typing error “CD3+ cells”
-Acronyms not defined: Please specify all acronyms reported.
The paper must also be revised for English fluency.
Author Response
Reviewer 2 Comments
Point 1: Abstract:
-Acronyms not defined: Please specify all acronyms reported.
Response 1:
HLA: Human Leucocyte Antigen
AML: Acute Myeloid Leukemia
MDS: Myelodysplastic Syndrome)
MDS – RAEB 2: Myelodysplastic Syndrome – Refractory Anemia with Excess Blasts 2
SAA: Severe Aplastic Anemia
Flu/Bu/CFA: Fludarabine/Busulfan/Cyclophosphamide
Mel/Flu: Melfalan/Fludarabine
Point 2: Introduction:
-Please revised typing error “CD3+ cells”
-Acronyms not defined: Please specify all acronyms reported.
Response 2:
SCID: Severe Combined Immunodeficiency Syndrome
CD34+ cells: Hematopoietic Stem Cells
CD3+ cells: T lymphocytes
GVHD: Graft Versus Host Disease
ATG: Antithymocite Globuline
NRM: Non-Relapse Mortality
RIC: Reduced Intensity Conditioning
CR1: Ffirst Complete Remission
G-CSF: Granulocyte colony-stimulating factor
TNC: Total Nucleated Cells

Reviewer 3 Report
The results presented in the manuscript entitled “Case series using salvage haplo-identical stem cells for secondary transplantation” are in a logical sequence to that contain data to inform the readers.
1) At first, some grammatical points can be seen in the text of the manuscript.
2) The novelty and hypothesis of the research must be involved at the end of the “Introduction section”.
3) The manuscript is not well referenced. All parts of the “Methods section” should be cited with related references.
4) Since the discussion section is one of the most important parts of the paper, this section must be improved with more attention and explanation. In the discussion section, results must be compared with another results from previous studies.
Some grammatical points can be seen in the text of the manuscript.
Author Response
Reviewer 1 Comments
Point 1: At first, some grammatical points can be seen in the text of the manuscript.
Response 1: See the new version of manuscript, please!
Point 2: The novelty and hypothesis of the research must be involved at the end of the “Introduction section”.
Response 2: Fundeni Clinical Institute is the only Bone Transplant Unit from Romania who perform haploidentical transplant procedures. This procedure can be also used as salvage option for those who had engraftment failure or rejection of the first stem cells graft.
Point 3: The manuscript is not well referenced. All parts of the “Methods section” should be cited with related references
Response 3: References for “Methods section” are listed below:
- Gaballa S, Ge I, El Fakih R, et al. Results of a 2-arm, phase 2 clinical trial using post-transplantation cyclophosphamide for the prevention of graft-versus host disease in haploidentical donor and mismatched unrelated donor hematopoietic stem cell transplantation. Cancer. 2016;122:3316–3326.
- Moiseev IS, Pirogova OV, Alyanski AL, et al. Risk-adapted GVHD prophylaxis with post-transplantation cyclophosphamide in adults after related, unrelated, and haploidentical transplantations. Eur J Haematol. 2018;100:395–402
- Bashey A, Zhang X, Sizemore CA, et al. T-cell-replete HLA-haploidentical hematopoietic transplantation for hematologic malignancies using post transplantation cyclophosphamide results in outcomes equivalent to those of contemporaneous HLA-matched related and unrelated donor transplantation. J Clin Oncol. 2013;31:1310–1316
- Yu CL, Zheng-Dong Qiao ZH, et al. The long-term outcome of reduced intensity allogeneic stem cell transplantation from a matched related or unrelated donor, or haploidentical family donor in patients with leukemia: a retrospective analysis of data from the China RIC Cooperative Group. Ann Hematol. 2017;96:279–288
- Sun Y, Beohou E, Labopin M, et al. Unmanipulated haploidentical versus matched unrelated donor allogeneic stem cell transplantation in adult patients with acute myelogenous leukemia in first remission: a retrospective pair-matched comparative study of the Beijing approach with the EBMT database. Haematologica. 2016;101:e352–e354.
- How J, Slade M, Vu K, et al. T cell-replete peripheral blood haploidentical hematopoietic cell transplantation with post-transplantation cyclophosphamide results in outcomes similar to transplantation from traditionally matched donors in active disease acute myeloid leukemia. Biol Blood Marrow Transplant. 2017;23:648–653
- Di Stasi A, Milton DR, Poon LM, et al. Similar transplantation outcomes for acute myeloid leukemia and myelodysplastic syndrome patients with haploidentical versus 10/10 human leukocyte antigen matched unrelated and related donors. Biol Blood Marrow Transplant. 2014;20:1975–1981
- Raiola AM, Dominietto A, di Grazia C, et al. Unmanipulated haploidentical transplants compared with other alternative donors and matched sibling grafts. Biol Blood Marrow Transplant. 2014;20:1573–1579.
- Rashidi A, Slade M, DiPersio JF, Westervelt P, Vij R, Romee R. Post-transplant high-dose cyclophosphamide after HLA-matched vs haploidentical hematopoietic cell transplantation for AML. Bone Marrow Transplant.2016;51:1561–1564.
- McCurdy SR, Kasamon YL, Kanakry CG, et al. Comparable composite endpoints after HLA-matched and HLA-haploidentical transplantation with post-transplantation cyclophosphamide. Haematologica. 2017;102:391–400.
- Bashey A, Zhang MJ, McCurdy SR, et al. Mobilized peripheral blood stem cells versus unstimulated bone marrow as a graft source for T-cell-replete haploidentical donor transplantation using post-transplant cyclophosphamide. J Clin Oncol. 2017;35:3002–3009.
- Ruggeri A, Labopin M, Bacigalupo A, et al. Bone marrow versus mobilized peripheral blood stem cells in haploidentical transplants using posttransplantation cyclophosphamide. Cancer. 2018;124:1428–1437.
- Bacigalupo A. Alternative donor transplants for severe aplastic anemia. Hematology Am Soc Hematol Educ Program. 2018;2018:467–473.
- Gyurkocza B, Sandmaier BM. Conditioning regimens for hematopoietic cell transplantation: one size does not fit all. Blood. 2014;124:344–353.
- Luznik L, O’Donnell PV, Symons HJ, et al. HLA-haploidentical bone marrow transplantation for hematologic malignancies using nonmyeloablative conditioning and high-dose, post transplantation cyclophosphamide. Biol Blood Marrow Transplant. 2008;14:641–650
- Baker M, Wang H, Rowley SD, et al. Comparative outcomes after haploidentical or unrelated donor bone marrow or blood stem cell transplantation in adult patients with hematological malignancies. Biol Blood Marrow Transplant. 2016;22:2047–2055.
- Cho BS, Yoon JH, Shin SH, et al. Comparison of allogeneic stem cell transplantation from familial-mismatched/haploidentical donors and from unrelated donors in adults with high-risk acute myelogenous leukemia.Biol Blood Marrow Transplant. 2012;18:1552–1563.
- Ciurea SO, Zhang MJ, Bacigalupo AA, et al. Haploidentical transplant with post transplant cyclophosphamide vs matched unrelated donor transplant for acute myeloid leukemia. Blood. 2015;126:1033–1040.
- Yu S, Fan Q, Sun J, et al. Haploidentical transplantation without in vitro Tcell depletion results in outcomes equivalent to those of contemporaneous matched sibling and unrelated donor transplantation for acute leukemia. Medicine (Baltimore). 2016;95:e2973.
- Tuve S, Gayoso J, Scheid C, Radke J, Kiani A, Serrano D, et al. Haploidentical bone marrow transplantation with post-grafting cyclophosphamide: multicenter experience with an alternative salvage strategy. Leukemia. 2011;25:880–3.
- Sugita J, Kawashima N, Fujisaki T, Kakihana K, Ota S, Matsuo K, et al. HLA-haploidentical peripheral blood stem cell transplantation with post-transplant cyclophosphamide after busulfan-containing reduced-intensity conditioning. Biol Blood Marrow Transplant.2015;21:1646–52
- Bashey A, Zhang M-J, McCurdy SR, St Martin A, Argall T, Anasetti C, et al. Mobilized peripheral blood stem cells versus unstimulated bone marrow as a graft source for T-cell-replete haploidentical donor transplantation using post-transplant cyclophosphamide. J Clin Oncol. 2017;35:3002–9
- Kasamon YL, Bolaños-Meade J, Prince GT, Tsai H-L, McCurdy SR, Kanakry JA, et al. Outcomes of nonmyeloablative HLA haploidentical blood or marrow transplantation with high-dose post-transplantation cyclophosphamide in older adults. J Clin Oncol. 2015;33:3152–61
- Kröger N, Iacobelli S, Franke G-N, Platzbecker U, Uddin R, Hübel K, et al. Dose-reduced versus standard conditioning followed by allogeneic stem-cell transplantation for patients with myelodysplastic syndrome: a prospective randomized phase III study of the EBMT (RICMAC Trial). J Clin Oncol. 2017;35:2157–64.
- Bornhauser M, Kienast J, Trenschel R, Burchert A, Hegenbart U, Stadler M, et al. Reduced-intensity conditioning versus standard conditioning before allogeneic haemopoietic cell transplantation in patients with acute myeloid leukaemia in first complete remission: a prospective, open-label randomised phase 3 trial.Lancet Oncol. 2012;13:1035–1044.
- Gorgeis J, Zhang X, Connor K, Brown S, Solomon SR, Morris LE, et al. T Cell-Replete HLA haploidentical donor transplantation with post-transplant cyclophosphamide is an effective salvage for patients relapsing after an HLA-matched related or matched unrelated donor transplantation. Biol Blood Marrow Transplant.2016;22:1861–6.
- Tischer J, Engel N, Fritsch S, Prevalsek D, Hubmann M, Schulz C,et al. Second haematopoietic SCT using HLA-haploidentical donors in patients with relapse of acute leukaemia after a first allogeneic transplantation. Bone Marrow Transplant. 2014;49:895–901.
Point 4: Since the discussion section is one of the most important parts of the paper, this section must be improved with more attention and explanation. In the discussion section, results must be compared with another results from previous studies.
Response 4: In the recent years, the haplo-identical family members became one of the valid choices for allogeneic stem cell transplant.
In Romania, the algorithm of choosing a donor for a patient starts with family HLA typing in order to:
- find a sibling donor as a first step
- additionally identify the haplotypes within the family in case of a haplo-identical transplant needed
The second step is to look out in international registries for a matched unrelated donor. In case of a patient who does not have sibling or matched unrelated donor in timely manner, the haplo-identical transplant is proposed.
We presented three cases using haplo protocol in order to save the lives of patients who had engraftment failure or rejections. The engraftment failure maybe due (in the first two cases) to the conditioning we used (Flu/Bu/CFA) combined with marrow grafts. The source of hematopoietic stem cells is also an important factor for failure of the hematopoietic stem cells (HSC) transplant. In all three cases the amount of CD34+ in the marrow graft was poor (0,97 x 106 CD34+cells/ kg recipient; 2,17 x 106 CD34+cells/ kg recipient; 3,1 x 106 CD34+cells/ kg recipient). It is well known that an insufficient quantity of stem cells in the graft can lead to engraftment failure. This small amount of stem cells from the bone marrow graft can be explained by the fact that, in all three cases, the weights of recipients were significantly higher than the weights of the donors. In addition, in the first two cases donors had a 50% HLA match with recipients.
For the second transplant, in all 3 patients was used another type of RIC conditioning (Mel/Flu) and another source of stem cells (peripheral blood stem cells). The amount of stem cells from peripheral blood was superior compared to the one from the first transplant (4,4 vs 0,97; 10,6 vs 2,17 respectively 5,23 vs 3,1). The donor for the third patient was changed for the second transplant, unlike the first two patients. This patient had a MUD 10/10 HLA match for the first transplant but for the second one, his haplo-identical sister with a 5/10 HLA match became the donor.
HLA-haploidentical stem cells post-transplant with high-dose cyclophosphamide is associated with low rate GVHD and non-relapse mortality (as Baltimore group has already demonstrated).
Both sources of stem cells grafts (BM, PBSC) are acceptable options in present time. All three patients grafted when peripheral blood was used as source of stem cells graft. This is in accordance with the medical literature that shows superior rates of engraftment for patients who receive peripheral blood stem cells (PBSC):
- Couban et al 2002: superior rates of engraftment for AML patients who receive peripheral blood stem cells (PBSC). In the same time, these patients have a higher risk for acute and chronic graft versus host disease (GVHD)
- Anaseti et al 2012: comparable survival rates and relapse rates in AML patients transplanted with PBSC MUD grafts vs. AML patients transplanted with bone marrow (BM) grafts. It is showed that BM grafts have a higher rates of graft failure and PBSC grafts are accompanied by higher rates of chronic GVHD.
- Schlenk et al 2010: MSD (matched sibling donor) and MUD (matched unrelated donor) are the preferred option for HSC-transplant. For patients who do not have these types of donors can be appealed haplo-identical donors.
In all three cases, the second transplant was haplo-identical peripheral blood stem cells using Mel/Flu conditioning; the cells engrafted properly and the patients became complete chimeras, two of them are alive with good quality of life.

Round 2
Reviewer 3 Report
All my comments have been implemented in the revised version of the manuscript.